# Visible and Near-Infrared Broadband Absorber Based on Ti_3_C_2_T_x_ MXene-Wu

**DOI:** 10.3390/nano12162753

**Published:** 2022-08-11

**Authors:** Yang Jia, Tong Wu, Guan Wang, Jijuan Jiang, Fengjuan Miao, Yachen Gao

**Affiliations:** 1Department of Optoelectronic Information, Electronic Engineering College, Heilongjiang University, Harbin 150080, China; 2College of Communication and Electronic Engineering, Qiqihar University, Qiqihar 161000, China

**Keywords:** MXene (Ti_3_C_2_T_x_), local surface plasmon, absorption, nanostructures, broadband, Fabry–Perot resonance

## Abstract

A high absorption broadband absorber based on MXene and tungsten nanospheres in visible and near-infrared bands is proposed. The absorber has a maximum absorption of 100% and an average absorption of 95% in the wavelength range of 400–2500 nm. The theoretical mechanism and parameter adjustability of the absorber are analyzed by FDTD solutions. The results show that the structural parameters can effectively adjust the absorption performance. The good absorption performance is due to the action of the local surface plasmon resonance coupling with the gap surface plasmon resonance and Fabry–Perot resonance. The simulation results show that the absorber is insensitive to the polarization and oblique incidence angle of incident light, and that high absorption and broadband can be maintained when the oblique incidence angle is up to 60°.

## 1. Introduction

With the development and improvement of nanophotonics, a large number of optical devices such as perfect lenses [1,2], polarization converters [3,4,5], absorbers [6,7,8], lasers [9], special light sources [10], etc. are designed. Absorbers, because of their application in the fields of solar energy conversion [11], infrared detection [12,13] and atmospheric environment monitoring [14], have attracted wide attention. How to improve the absorption performance of absorbers has become the primary problem faced by scholars.

In recent years, the discovery of metamaterials has broken through peoples’ understanding of optical control devices, and a large number of optical devices with various functions have been designed and manufactured. Among them, metamaterial absorbers show irreplaceable advantages in various application fields. In 2018, Chen et al. [15] designed a broadband absorber based on gold nanowires in ultraviolet band and visible band, and the absorbance of the absorber could reach 90%. In 2020, Jiang et al. [16] studied a broadband solar absorber based on hyperbolic metamaterial using photonic topological transitions (PTTs), and an absorbance of up to 90% with the vertical incident light in the wavelength range of 300–2200 nm could be achieved. In addition, applying refractory metals (tungsten (W), chromium (Cr), nickel (Ni), titanium nitride (TiN), zirconium nitride (ZrN), etc.) metamaterials, a series of absorbers with good performance have also been put forward. In 2020, Bilal et al. [17] proposed a metamaterial absorber based on tungsten elliptical rings; the proposed absorber has an absorbance of more than 90% in the visible band. In 2021, Hou et al. [18] studied a broadband metasurface absorber using TiN with the method of the finite difference time domain (FDTD) and obtained a good performance which has an average absorbance up to 95%. In 2021, Bilal et al. [19] designed a concentric triangular ring metasurface absorber based on chromium metal, which is insensitive to the angle of incidence and has an absorbance of more than 90% in the visible band. Utilizing refractory metals, such metamaterial absorbers show high temperature tolerance, and their broadband absorption performance also makes them have a good application prospect in solar energy harvest, photovoltaic and other fields.

Two-dimensional (2D) materials [20,21,22] have ultra-high carrier mobility and unique size advantages that make them stand out among many materials. Graphene [23,24], the world’s first 2D material, has been studied in various fields for its strong electric field constraint ability, low loss, good heat conduction performance and adjustable band gap. Recently, a new class of quasi-2D materials, MXene [25,26,27,28,29], a class of 2D inorganic compounds, consisting of carbides, carbonitrides, and transition metals, has come into focus and sparked a new wave of research enthusiasm. The molecular formula for the MXene group is written as M_n+1_X_n_T_x_, where M is for transition metal, X stands for C or N and T stands for functional groups (-OH, -O, -F) [30]. Due to the larger specific surface area, better stability, unique dielectric multilayer structure and semiconductor properties, Ti_3_C_2_T_x_ MXene demonstrates excellent electrical and optical properties, which makes it a good choice for manufacturing optical devices, especially for absorbers, and more and more scholars show intense interest in it. Meanwhile, their abundant surface functional groups, 2D laminar structure similar to potato chips, and thickness-dependent electrical and optical properties also provide a good prospect for the further improvement of the device performance [31,32]. At present, single or multilayer Ti_3_C_2_T_x_ MXene can be easily obtained by wet chemical etching [33], and it has been applied to ion battery [34], electromagnetic shielding [35], absorber [36] and other aspects. Especially in the application of a perfect absorber, Ti_3_C_2_T_x_ MXene has an irreplaceable effect. In 2017, Ahmed et al. [37] studied the preparation and related performance of the anode of lithium ion battery based on 2D titanium carbide. The cycling performance of the anode can be greatly improved by the deposition of SnO_2_ and HfO_2_ passivation layer on the atomic layer. In 2018, Choi et al. [38] applied an MXene colloidal solution to the slot antenna array, increasing the terahertz transmission shielding performance by three orders of magnitude. In 2018, Krishnakali et al. [39] designed a broadband plasma absorber based on the MXene cylindrical array with an absorption efficiency of 85% within the range of 600 to 2100 nm. In 2020, Ali et al. [40] designed a broadband wide-angle absorber with MIM structure based on MXene and gold nanoparticles, which realized angle independent near-perfect absorption from ultraviolet to near-infrared bands. In 2020, Guo et al. [41] prepared an ultra-thin microwave absorber with MXene/SiO_2_ core edge structure by utilizing the rich functional groups around MXene. By adjusting the thickness of the SiO_2_ layer, the reflection loss was lower than −20 dB in the X-band and Ku band, and the maximum absorption efficiency was greater than 99%. In 2021, Cui et al. [42] used MXene combined with gold nanorods to design an absorber in visible and near-infrared wavelengths, and the local maximum absorption efficiency reached 99% by tuning the model parameters. Although the application of Mxene in many fields has achieved some progress, how to use MXene to develop a broadband absorber is still an urgent problem to be solved, which is also of great significance to the development of photic devices.

In this work, a broadband absorber based on MXene (Ti_3_C_2_T_x_) and tungsten nanospheres was proposed. The absorption characteristics of the absorber were studied by using FDTD Solutions software. It was found that the designed absorber has an ultra-broadband absorption characteristic in the visible and near-infrared band, and the absorber is insensitive to the angle of incident light. Based on the analysis of electric field distribution, the mechanism of broadband absorption was discussed theoretically. The proposed absorber displays great stability and excellent absorption characteristics, and it has potential application prospects in the fields of solar energy harvest, seawater evaporation, photovoltaic, etc.

The rest of this paper is elaborated in the following aspects. In Section 2, the structure and the materials of the absorber are provided, and the parameter setting of the simulation software is also presented. The main results and the discussions are given in Section 3. Ultimately, the conclusion is reached in Section 4.

## 2. Structure and Simulation Model

The structure of the broadband absorber is shown in Figure 1. As is shown in Figure 1a, the absorber consists of four layers, which are the bottom tungsten layer, MXene layer, PMMA layer and tungsten nanosphere layer with the thickness of *L*_1_ = 400 nm, *L*_2_ = 60 nm, *L*_3_ = 200 nm, and the nanosphere diameter of *D* = 34 nm. The tungsten nanospheres layer is composed of periodically arranged tungsten nanospheres closely. The whole structure is on the SiO_2_ substrate. At the same time, in order to prevent the oxidation of tungsten nanospheres exposed to the air for a long time, another layer of PMMA protective layer with a thickness of 123 nm is set on the nanosphere layer. Figure 1b shows the cross-sectional view of the absorber structure at the center of nanospheres.

The refractive coefficients of metal tungsten and SiO_2_ follow reference [38]. The refractive index of PMMA is set to be 1.4813. The Drude–Lorenz model is used to fit the permittivity of MXene adopted in this paper [42].
(1)$$\varepsilon_{Drude}=\varepsilon_{1}-{\left(\frac{\omega_{p}}{\omega }\right)}^{2}+i\left(\varepsilon_{2}-\frac{\omega_{p}^{2}\gamma }{\omega^{3}+\omega \gamma^{2}}\right)$$
(2)$$\varepsilon_{Lorenz}=\varepsilon_{3}\left[1+\frac{\omega_{p}^{2}\left(\omega_{0}^{2}-\omega^{2}\right)+i\omega \omega_{0}^{2}\gamma }{{\left(\omega_{0}^{2}-\omega^{2}\right)}^{2}+\omega^{2}\gamma^{2}}\right]$$
where the plasma frequency $\omega_{p}=4.21\times 10^{14}\mathrm{rad}/s$, scattering losses $\gamma =8.62\times 10^{14}\mathrm{rad}/s$, Lorentz pole frequency $\omega_{0}=2.3\times 10^{15}\mathrm{rad}/s$, $\varepsilon_{1}=6$, $\varepsilon_{2}=3$, $\varepsilon_{3}=0.2$. The fitting curve of the permittivity for MXene is shown in Figure 2 [39].

In order to analyze the performance of the designed absorber, the 3D finite-difference time-domain (FDTD) method is used (software: FDTD Solutions). During the simulation, the boundary conditions in the X direction and Y direction are set as the periodic boundary, and the Z direction is set as the perfect matching layer. A plane wave with X-polarized is incident along the negative Z-axis, and the incident light satisfies the following Maxwell wave equation [40]:(3)$$\nabla \times \frac{1}{\mu_{r}}\left(\nabla \times E\right)-K_{0}^{2}\left(\varepsilon_{r}-\frac{j\sigma }{\omega \varepsilon_{0}}\right)E=0$$
where $\mu_{r}$ is the relative permittivity, $\varepsilon_{r}$ is the permeability, K is the propagating vector of the wave defined for all mediums in the model, ω is the angular frequency, and σ is the conductivity of the medium. The reflection *R* and transmission *T* of the absorber are calculated by using the boundary reflection and transmission coefficient, so the absorption can be calculated according to the formula *A =* 1 *– R − T*. In order to prevent transmission, the thickness of the bottom tungsten layer is set as *L*_3_ = 200 nm throughout the work, which is much larger than the skin depth, so the transmission *T* can be seen as 0, and the absorption can be rewritten as *A =* 1 *− R* [43,44,45,46].

## 3. Results and Discussions

### 3.1. Absorption Efficiency of the Proposed Absorber and Derived Absorbers

Under the vertical incidence of X-polarized light, the reflection and absorption spectrum of the proposed absorber are obtained and shown in Figure 3. As can be seen from Figure 3, there are two absorption valleys at 909 nm and 1504 nm with the absorption efficiency of 98.2% and 97.8%, three absorption peaks at 627 nm, 1189 nm and 1735 nm, with the corresponding absorption efficiency 99.7%, 100% and 99%, respectively. Thus, two absorption valleys and three absorption peaks together constitute a broadband absorption spectrum in visible and near-infrared ranges. The absorption bandwidth with the absorption over 80% is 1955 nm. The absorption spectra obtained has a wavelength range of 1747 nm with the absorption more than 90%, and 1570 nm with the absorption even more than 95%.

In addition, in order to prove that the tungsten nanospheres used in the absorber are superior to other forms of nanoparticles, we replace the tungsten nanospheres by tungsten nanorods with the same diameter and keep other parameters unchanged. Figure 4 shows the absorption curve of the tungsten nanorods absorber, in which the ratio of long diameter to short diameter (RLS) of the nanorods increases from 1 to 2.2. It is worth noting that when the RLS of the nanorods is 1, the nanorod is morphologically similar to a nanosphere. As can be seen from Figure 4, with the increase in RLS, the overall absorbance of the absorber gradually weakens, especially in the near-infrared band, which is due to the fact that tungsten nanospheres and nanorods have different plasmon resonance frequencies, and the shape characteristics of the nanospheres and the nanorods will make nanospheres generate one plasma resonance absorption peak while the nanorods generate two under the same incident light. Moreover, according to the law of energy conservation, the plasmon resonance strength of the nanorods is always weaker than that of the nanospheres. Therefore, tungsten nanospheres are chosen as the main nanoparticle for the absorber design.

In order to clearly understand the origin of high efficiency broadband absorption of the designed absorber, the MXene/tungsten-based absorbers without the tungsten nanosphere layer and without the MXene layer are also studied. Figure 5 shows the absorption spectra of the proposed absorber with the tungsten nanosphere array absent (Figure 5a) and the Mxene layer absent (Figure 5b), respectively.

From Figure 5a, it can be seen that in the case of absence of tungsten nanospheres, there are two absorption peaks at 450 nm and 1030 nm, and the maximum absorption efficiency can reach 97%. However, the absorption efficiency decreases significantly in the range of near-infrared, and the absorption efficiency is only 26% at 2500 nm. The result in Figure 5a is related to the properties of Mxene itself, and such an absorption spectrum is mainly attributed to the interband and intraband transitions of electrons, which is consistent with the results in reference [40]. As shown in Figure 5b, with the absence of an MXene layer, the maximum absorption efficiency, which peaks at 536 nm and 1240 nm, is 90% and 82%, respectively. Similarly, in the near-infrared region, the absorption spectrum drops sharply and the absorption is only 10.2% at 2500 nm. The high absorption at shortwave range is due to the local surface plasmon resonance (LSPR) around the tungsten nanospheres and the coupling of LSPR of adjacent nanospheres, which is highly consistent with the results in reference [43].

### 3.2. Size Effects of the Absorber

In this section, we will study how the performance of the absorber changes with the thickness of MXene layer *L*_1_*,* the distance between the nanosphere layer and MXene layer (the thickness of PMMA layer) *L*_2_, the diameter of tungsten nanosphere *D* and the distance *d* between tungsten nanospheres, respectively.

Firstly, the effect of the distance *L*_2_ between tungsten nanospheres (center of sphere) and MXene layer on the absorption performance is studied. The nanospheres are packed tightly together by *d* = 0 nm and the diameter *D* = 34 nm. The thickness of the MXene layer is *L*_1_ = 30 nm (Figure 6a) and *L*_1_ = 400 nm (Figure 6b). Figure 6a,b show the absorption curves of the absorber with *L*_2_ increasing from 40 to 130 nm.

From Figure 6a,b, it can be clearly seen that whether the thickness of the MXene layer is 30 nm (Figure 6a) or 400 nm (Figure 6b), the three absorption peaks all appear at the same points, which is at 680 nm, 1420 nm and 2200 nm, and they together constitute the highly efficient and broadband absorption spectrum. At the same time, with *L*_2_ increasing from 40 to 130 nm, the overall average absorption increases first, reaches the maximum when *L*_2_ = 60 nm, and then decreases gradually. In particular, in the band of 600–900 nm, with the increase in *L*_2_, the absorption first remains unchanged and then decreases faster and faster, and the absorption peaks at 680 nm will have a blue shift. In the band of 1600–2500 nm, the absorption first increases and then decreases after *L*_2_ increases to 80 nm, also, the absorption peaks at 2200 nm will have a red shift. The absorption peak at 1420 nm changes very slightly, which is the same as the trend of the peak at 2200 nm. We speculate that the main reason for the situation in the visible band is due to the LSPR and the gap surface plasmon (GSP) resonance between the MXene layer and tungsten nanosphere layer [42]. The variation of absorption peaks in the near-infrared band can be attributed to Fabry–Perot (FP) resonance [30,31,40]. It is worth noting that, as is shown in Figure 5b, when *L*_2_ = 40 nm with the wavelength of incident light 944–1269 nm, *L*_2_ = 50 nm with the wavelength of incident light 769–1349 nm, *L*_2_ = 60 nm with the wavelength of incident light 766–1382 nm and *L*_2_ = 70 nm with the wavelength range of incident light 1298–1382 nm, the absorption even reaches 100%.

Next, we demonstrate the effect of the diameter *D* of tungsten nanospheres on the performance of the absorber. The nanospheres are packed tightly together by *d* = 0 nm and the distance between the tungsten nanospheres (center of sphere) and MXene layer is *L*_2_ = 90 nm, the thickness of the MXene layer is *L*_1_ = 30 nm (Figure 7a) and *L*_1_ = 400 nm (Figure 7b). Figure 7 shows the absorption curves of tungsten nanospheres with the diameter *D* of nanospheres changing from 10 to 46 nm.

It is obvious that in both Figure 7a,b, the overall absorption performance is enhanced with the increase in the diameter of the nanospheres, and the absorption performance is optimal when *D* = 34 nm and then gradually decreases. Here, the absorption spectrum only has two absorption peaks at 590 nm and 1420 nm, and with the increase of *D*, they both have red shifts. It is important to note here that when *D* increases to 26 nm, a new absorption peak appears at 2200 nm as in Figure 7a,b, and the peak will disappear again when diameter *D* exceeds 38 nm. The main reason for this situation is attributed to the indirect changes in the FP cavity and the destruction of the GSP. Similarly, in Figure 5b, we also observe that when the diameter of the nanospheres is 22 nm with the wavelength of the incident light 1417–1789 nm, and the diameter of the nanospheres is 26 nm with the wavelength of the incident light 1371–1639 nm, the absorption reaches 100% once again.

Finally, the influence of the distance *d* between nanospheres on the absorption performance of the absorber is studied. The relevant parameters of the absorber are set as *L*_2_ = 90 nm, *D* = 34 nm, *L*_1_ = 30 nm (Figure 8a) and *L*_1_ = 400 nm (Figure 8b). Figure 8a,b shows the absorption curves of the absorber with the distance *d* between nanospheres increasing from 0 to 10 nm.

From Figure 8a,b, it can be seen that with the increase in *d*, the average absorption efficiency will decrease, and the absorption increases slightly only around 500 nm. The three absorption peaks (680 nm, 1420 nm, 2200 nm) gradually become two (600 nm, 1350 nm) as *d* increases. The absorption peak at 1350 nm will also have a blue shift with the increase in *d*.

From Figure 6a,b, Figure 7a,b, Figure 8a,b, it is obvious that the thickness of Mxene has an influence on the absorption performance. Compared with 30 nm Mxene, the absorption performance of the 400 nm Mxene layer has a slight improvement. This is related to the properties of MXene. Compared with 30 nm MXene, 400 nm thick MXene has a larger imaginary part of permittivity, as is shown in Figure 2b, which also leads to a larger energy loss, making the absorption at 400 nm higher than that at 30 nm. The power loss density is calculated as follows [44]:(4)Ploss=12ωImεE2
where E is the electric field through MXene, ω is the wavelength, and the Imε is the permittivity of the MXene layer.

### 3.3. Physical Mechanism Analysis of the Absorber

#### 3.3.1. Impedance Matching Theory Analysis

In order to theoretically clarify the physical mechanism of broadband absorption of the absorber, impedance matching theory is introduced. It has been shown in Section 2 that the bottom of the proposed absorber is formed by a sufficiently thick tungsten film, whose thickness is greater than the skin depth of incident light. So, the transmittance of the absorber is $T=S_{12}=0$. The effective impedance can be calculated by the following formula [17,44]:(5)$$Z_{r}=\sqrt{\frac{{\left(1+S_{11}\right)}^{2}-S_{12}^{2}}{{\left(1-S_{11}\right)}^{2}-S_{12}^{2}}}=\frac{1+S_{11}}{1-S_{11}}$$
(6)$$A=1-R=1-\left|\frac{Z-Z_{0}}{Z+Z_{0}}\right|=1-{\left|\frac{Z_{r}-1}{Z_{r}+1}\right|}^{2}$$
where $Z_{r}=\frac{Z}{Z_{0}}$ is the normalized complex impedance, Z and $Z_{0}$ are the effective impedance and free space impedance, and $S_{11}$ and $S_{12}$ are the reflectance and transmittance calculated by S-parameters. As is shown in Formula (6), when $Z_{r}=1$, that is to say, the effective impedance matches the free-space impedance perfectly, the absorbance of the absorber is the maximum. Figure 9a,b show the real and imaginary parts of normalized complex impedance, respectively. As is shown in Figure 9a,b, normalized complex impedance has a real part equal to 1 and an imaginary part equal to 0 at the wavelength around 627 nm, 1189 nm and 1735 nm, and in the range of 560–2390 nm, the real part is always close to 1. This means that the impedance of the absorber at the absorption peak matches closely the impedance of free space. Therefore, the designed absorber has a high broadband absorption in the range of visible and near-infrared band.

#### 3.3.2. The Electric Field Analysis

In order to further explore the physical mechanism of high absorption and broadband effect of the absorber, the electric field at the resonance frequency is plotted in Figure 10. The plane wave is incident along the *Z*-axis. Figure 10a–c show the electric field distribution of the absorber with wavelength of 627 nm, 1189 nm and 1735 nm on the plane X = 0 nm, respectively. Figure 10d depicts the electric field distribution of tungsten nanosphere in the Z = 60 nm plane.

It is obvious from Figure 10a–c that the electric field is bound around the tungsten nanosphere and between the nanosphere and the MXene layer, and the local electric field at 627 nm and 1189 nm is stronger than that at 1735 nm. This is the main factor of broadband absorption. As is shown in Figure 10d, the electric field is enhanced on both sides of the nanosphere, which proves the existence of the LSPR. All of these demonstrate that the high-efficiency broadband absorption is due to the existence of LSPR, GSP and FP resonance [42,47].

Up to this point, the parameter dependence of the absorption can be explained more clearly. The high efficiency broadband absorption exhibited by the designed absorber is attributed to the combination of LSPR, GSP and FP resonance. The LSPR is generated around the tungsten nanospheres and is coupled between adjacent nanospheres. According to Figure 5b, LSPR enables the absorber to have an absorption to a certain extent at short wavelength range. Moreover, the GSP is formed when the resonators are placed near the metal surface but separated by a nanoscale gap. It occurs due to the interaction of surface plasmons on adjacent dielectric-metal surfaces and the reflection of adjacent surfaces of the resonator (the nanosphere). The absorber designed in this paper meets the conditions of GSP by tightly arranged tungsten nanospheres, the MXene layer and the PMMA layer in the middle. The GSP generates standing wave resonance in the gap between the MXene layer and the nanosphere layer, thus further improving the absorption efficiency at the short wavelength range. In the near-infrared region, the tungsten nanospheres, MXene layer and PMMA layer formed a similar metal–dielectric–metal (MDM) structure, a Fabry–Perot-like cavity, resulting in Fabry–Perot (FP) resonance [30,31,40], and thus, the electric field constraint is formed and the energy consumption is increased, which broadens the absorption bandwidth to cover the entire near-infrared region. Based on the above foundation, the final high efficiency broadband absorber is formed. However, when *L*_2_ and *D* change, the strength of the LSPR and the density of gap between the MXene layer and the nanosphere layer are directly or indirectly changed, causing the variation of GSP and FP resonance. As shown in Figure 6a, with the increase in *L*_2_, the gap of GSP cavity increases, leading to the visible absorption decline. At the same time, the change of FP cavity mode density leads to the trend of FP resonance strengthening first and then weakening. This trend is generally evident in Figure 6. When the nanosphere spacing starts to increase, the FP cavity is completely destroyed, and with the increase in the gap, more light is reflected back from the cracks, and therefore, the absorption goes down in the near-infrared region.

### 3.4. Robustness of Polarization Angle and Incident Angle

In the practical application of the absorber, the incident light cannot be always incident vertically to the surface of the absorber or polarized in the X direction, so it is very important to study the stability of the absorber when the incident light has an oblique angle and the polarization angle is not 0. As is shown in Figure 11a,b, the parameters of the absorber are set as *L*_1_ = 400 nm, *L*_2_ = 90 nm, *D* = 34 nm, and *d* = 0, and the incident light polarizes along the X-direction. Figure 11a shows the absorption spectrum with the incident angle increasing from 0° to 60° and Figure 11b shows the absorption spectrum with the polarization angle increasing from 0° to 90°. From Figure 11a, as the angle increases to 50°, the absorption in the range of 400–610 nm and 2250–2500 nm decreases but remains above 70%. When the angle increases up to 60°, the absorption in the range of 2350–2500 nm decreases to 65%, but it still retains the high and broadband absorption efficiency in the range of 620–2150 nm. Therefore, the designed absorber can achieve the absorption performance of a larger incidence angle. From Figure 11b, It can be clearly seen that when the polarization angle of incident light increases from 0° to 90°, the absorption spectrum is almost unchanged, and the structure of the absorber is diagonally symmetric, indicating that the absorber is not sensitive to the polarization of incident light.

MXene, as a new quasi-2D material, has been applied to the absorber. For comparison, Table 1 shows the results of recent broadband absorbers.

In reference [5], a polarization sensitive and tunable grating structure with high absorption is designed in the 500–1200 nm range. In references [39,42], the absorption efficiency of the designed absorbers can reach more than 80%, however, the absorption efficiency of the two kinds of absorbers will decrease to some extent in the near-infrared region. In addition, reference [39] needs to etch MXene into a nanocolumn array to achieve the desired effect. In reference [43,47], the efficiency of the absorbers is limited by the number of structural layers. In reference [43], more than 90% absorption can be obtained by constructing nanoparticle structures with more than eight tungsten nanosphere layers. In reference [45], by using the thermal tunability of refractive index of VO_2_ material, a petal-shaped absorber with absorption efficiency and wavelength temperature-controlled is designed. The absorber designed in this paper is composed of patternless MXene combined with tungsten nanospheres, which is easy to realize in the manufacturing process and has an extremely broad bandwidth, covering the whole visible and near-infrared regions, and it has high absorption performance. By adjusting the structure parameters, the average absorbance can reach 95%; especially in some ranges, it can achieve perfect 100% absorption.

For the manufacturing process of the designed absorber, firstly, a 200 nm tungsten metal film is plated on the SiO_2_ substrate with a coating process. Then, the colloidal solution of MXene film can be easily obtained by a minimally intensive layer delamination method (MILD) [40], and the MXene colloidal solution is spin-coated on the tungsten metal film to obtain the MXene film. Different thickness MXene films can be obtained by varying the spin-coater speed and adjusting the time of the revolution. Next, the MXene film is coated with another layer of PMMA, and the Wu nanosphere layer is produced by a self-assembly process. Finally, another layer of PMMA is spin-coated to enhance the overall absorbance of the absorber and protect the tungsten nanosphere layer at the same time.

## 4. Conclusions

In summary, a high efficiency broadband absorber in visible and near-infrared bands is proposed, which is mainly composed of a patternless Ti_3_C_2_T_x_ MXene layer, refractory tungsten nanosphere layer and PMMA layer. By optimizing the structural parameters of the absorber, over 95% average absorption can be obtained in the wavelength range from 400 to 2500 nm. The FDTD method is used to study the theoretical mechanism and parameter dependence of high absorption and broadband. The results show that the high absorption and broadband are mainly caused by the LSPR of tungsten nanoparticles, GSP and FP resonance between Mxene and tungsten nanoparticles. In addition, the absorber is insensitive to the polarization and incidence angle of incident light, and it can still maintain high absorption and broadband performance at the incident angle of 60°. The absorption performance of the absorber is better than most reported solar absorbers, which provides a new idea for the application of solar energy harvest, solar water evaporation and other fields.

## Figures and Tables

**Figure 1 nanomaterials-12-02753-f001:**
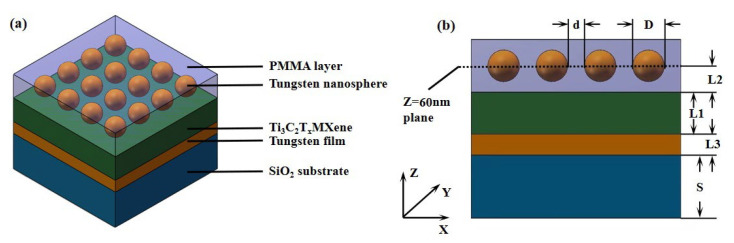
(**a**) Structure diagram of MXene/Wu-based absorber. (**b**) The cross-sectional view at the center of any row of nanospheres.

**Figure 2 nanomaterials-12-02753-f002:**
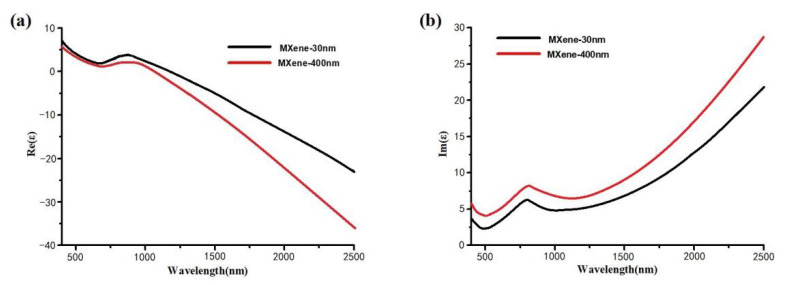
The fitting curve of the permittivity for MXene: (**a**) the real part; (**b**) the imaginary part.

**Figure 3 nanomaterials-12-02753-f003:**
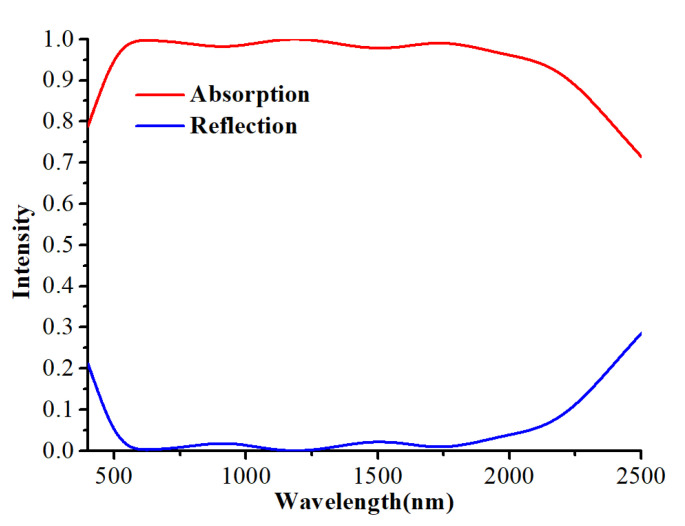
Absorption and reflection spectra under x-polarized plane waves.

**Figure 4 nanomaterials-12-02753-f004:**
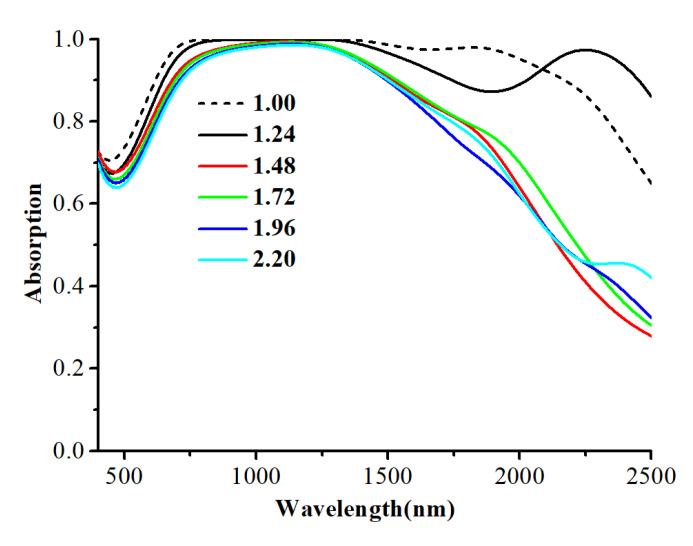
The absorption curve of the tungsten nanorods absorber with different RLS.

**Figure 5 nanomaterials-12-02753-f005:**
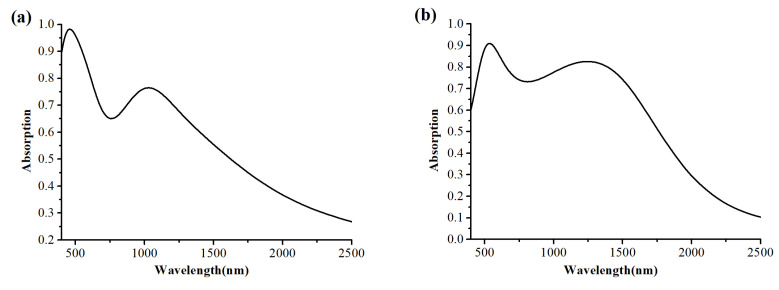
The absorption spectra of the proposed absorber: (**a**) tungsten nanosphere array is absent; (**b**) Mxene layer is absent.

**Figure 6 nanomaterials-12-02753-f006:**
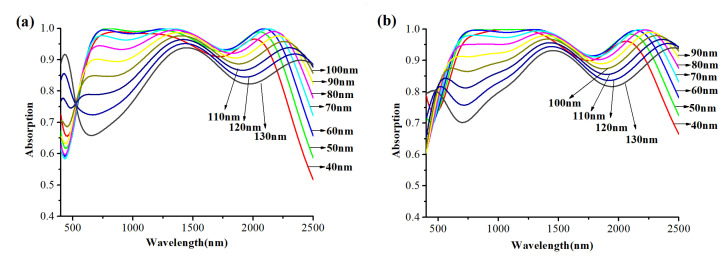
The absorption curves of the absorber with the distance *L*_2_ between tungsten nanosphere layer and MXene layer increasing from 40 to 130 nm: (**a**) the thickness of the MXene layer is 30 nm; (**b**) the thickness of the MXene layer is 400 nm.

**Figure 7 nanomaterials-12-02753-f007:**
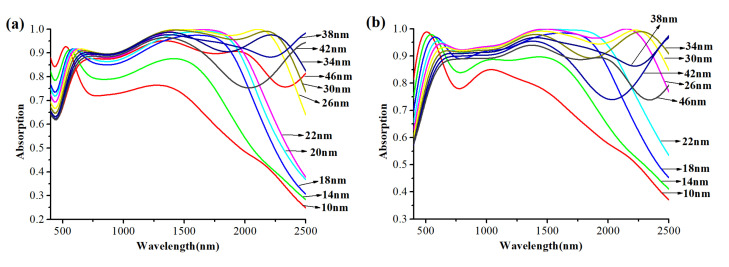
The absorption curves of the absorber with the diameter of the tungsten nanosphere increasing from 10 to 46 nm: (**a**) the thickness of the MXene layer is 30 nm; (**b**) the thickness of the MXene layer is 400 nm.

**Figure 8 nanomaterials-12-02753-f008:**
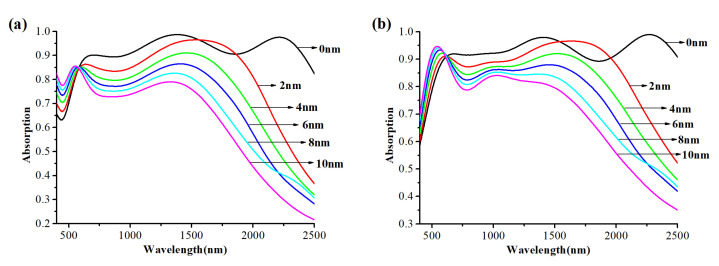
The absorption curves of the absorber with the distance *d* between nanospheres increasing from 0 to 10 nm: (**a**) the thickness of the MXene layer is 30 nm; (**b**) the thickness of the MXene layer is 400 nm.

**Figure 9 nanomaterials-12-02753-f009:**
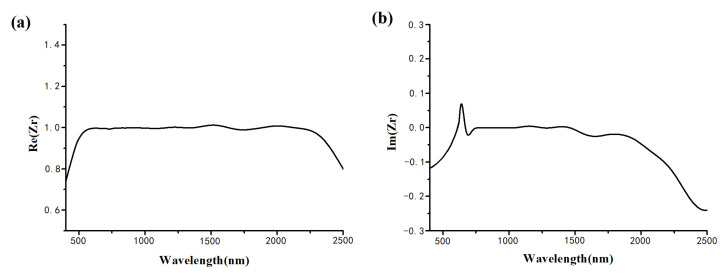
The real part (**a**) and imaginary part (**b**) of normalized complex impedance $Z_{r}$ of the absorber.

**Figure 10 nanomaterials-12-02753-f010:**
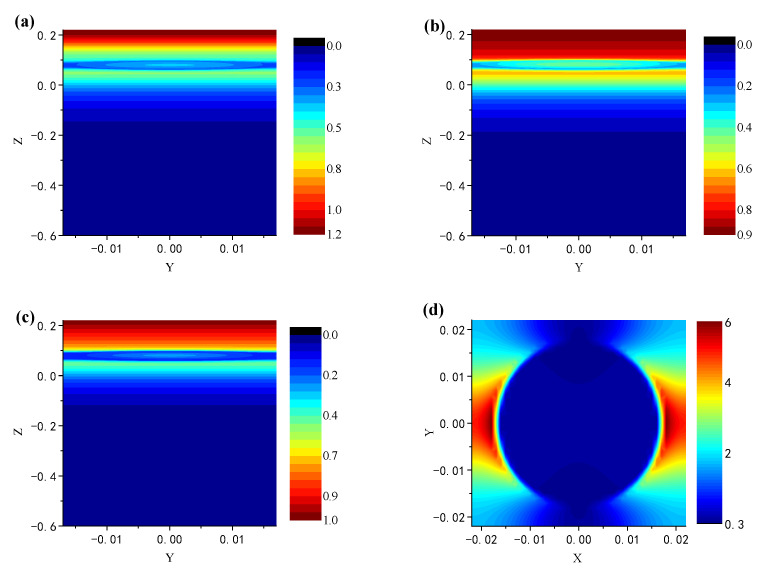
Electric field distribution of MXene/tungsten-based absorber: The electric field distribution at wavelength (**a**) 627 nm, (**b**) 1189 nm, and (**c**) 1735 nm on the X = 0 plane; (**d**) electric field distributed on the Z = −60 nm plane when the wavelength is 627 nm.

**Figure 11 nanomaterials-12-02753-f011:**
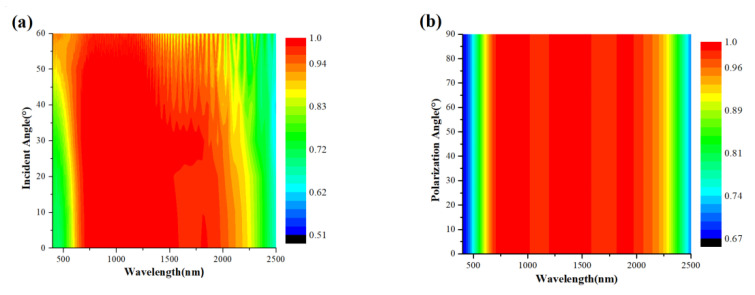
Absorption of MXene/tungsten-based absorber at (**a**) different incidence angles and (**b**) different polarization angles.

**Table 1 nanomaterials-12-02753-t001:** Comparison of the absorber in this paper with the published broadband absorbers in visible and near-infrared bands.

Reference	Main Material	Absorption Bandwidth	Absorption Efficiency
[5]	Au and Cr	500–1200 nm	90%
[39]	MXene, Au, and Al_2_O_3_	600–2100 nm	80–90%
[42]	MXene and Au	500–1500 nm	about 80%
[43]	Tungsten	500–2500 nm	90%
[45]	SiO_2_-VO_2_-MoS_2_	800–2350 nm/800–1160 nm	86.5%/96.6%
[47]	ITO	1500–3500 nm	over 80%
This paper	MXene and Tungsten	400–2500 nm	over 95%

## Data Availability

The data are included in the main text.

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
