# Peer review of "Visible and Near-Infrared Broadband Absorber Based on Ti3C2Tx MXene-Wu"

_nanomaterials, 2022, doi:10.3390/nano12162753_

Round 1

Reviewer 1 Report

The article titled, “Visible and near-infrared broadband absorber based on Ti3C2TxMXene-Wu” has been submitted by Gao and coauthors in Nanomaterials. In this article, a high absorption broadband absorber based on MXene and tungsten nanospheres in visible and near-infrared bands is proposed. The absorber has a maximum absorption of 100% and an average absorption of 95% in the wavelength range of 400nm-2500nm. The theoretical mechanism and parameter adjustability of the absorber are analyzed by FDTD. The simulation results show that the absorber is insensitive to the polarization and oblique incidence angle of the incident light and that high absorption and broadband can be maintained when the oblique incidence angle is up to 60°.

Overall, the article is well written and comprises some interesting facts. However, it can be further improved. I have some comments and suggestions. The authors are requested to address them point by point.

1.     MXene is quite a new kind of 2D material. What is the motivation behind using this material? It is requested to highlight the advantages of using this material and mention its superiority over other available materials. A comparative analysis will be a good demonstration.

2.     How can this proposed absorber be fabricated? What are the main challenges and possible limitations during the fabrication of this Nanoscale device? Though it is a simulation study, still there is a need to address these challenges. Please check.

3.     The authors should also discuss the possible applications of this proposed absorber related to solar energy harvesting and STPV etc. in the introduction because they mentioned in various places that it will be useful for these applications.

4.     On line number 215, the authors mentioned that improvement in the absorption is due to the properties of MXenes. There is no explanation provided that which properties are basically contributing to improving the absorption. Please it needs further clarification.  

5.     The authors should also discuss the class of metamaterial-based absorbers which used other plasmonic and refractory metals including tungsten (W), chromium (Cr), Nickel (Ni), TiN, ZrN, etc. Please check some of the important and useful articles, https://www.nature.com/articles/s41598-020-71032-8; https://opg.optica.org/josaa/abstract.cfm?uri=josaa-39-1-136)

6.     Impedance matching theory is the prerequisite to calculating the absorption of metamaterial-based absorbers and it seems missing in the article. It is suggested to include the impedance matching theory in the appropriate location of the article. Previously suggested articles will be helpful in this regard.  

7.     The authors used permittivity curves of MXenes during their simulation. It is requested to please provide related references.

8.     Fabry-Perot resonance is quite a critical factor in this type of absorber. So, I recommend creating a separate section for this and discussing in detail this phenomenon. 

9.     I believe that figures quality can be further improved. Please check the previously mentioned articles. Also, please remove the typos like spacing, font, units, etc., in the article.

Reviewer 2 Report

Authors propose a  high absorption broadband absorber based on Mxene and tungsten nanospheres in visible and near-infrared bands. My proposed changes are as follows:

1.      Authors should justify the usage of Tungsten nanospheres. Is it possible to replace them with nanorods?

2.      Authors are missing some recent articles in the field such as Investigation of Hyperbolic Metamaterials.

3.      It would be desirable to compare the obtained results with the experimental outputs.

4.      To allow for a better readability, Authors should add a separate paragraph in the Introduction describing composition of the manuscript in terms of the proposed Sections.

Round 2

Reviewer 1 Report

Yes, the article has been substantially improved and can be accepted for the publication. There is no further revision required.